# Convolutional Neural Networks on non-uniform geometrical signals using Euclidean spectral transformation

**Chiyu "Max" Jiang**
UC Berkeley

**Dequan Wang**
UC Berkeley

**Jingwei Huang**
Stanford University

**Philip Marcus**
UC Berkeley

**Matthias Nießner**
Technical University of Munich

## Abstract

Convolutional Neural Networks (CNN) have been successful in processing data signals that are uniformly sampled in the spatial domain (e.g., images). However, most data signals do not natively exist on a grid, and in the process of being sampled onto a uniform physical grid suffer significant aliasing error and information loss. Moreover, signals can exist in different topological structures as, for example, points, lines, surfaces and volumes. It has been challenging to analyze signals with mixed topologies (for example, point cloud with surface mesh). To this end, we develop mathematical formulations for Non-Uniform Fourier Transforms (NUFT) to directly, and optimally, sample nonuniform data signals of different topologies defined on a simplex mesh into the spectral domain with no spatial sampling error. The spectral transform is performed in the Euclidean space, which removes the translation ambiguity from works on the graph spectrum. Our representation has four distinct advantages: (1) the process causes no spatial sampling error during the initial sampling, (2) the generality of this approach provides a unified framework for using CNNs to analyze signals of mixed topologies, (3) it allows us to leverage state-of-the-art backbone CNN architectures for effective learning without having to design a particular architecture for a particular data structure in an ad-hoc fashion, and (4) the representation allows weighted meshes where each element has a different weight (i.e., texture) indicating local properties. We achieve results on par with the state-of-the-art for the 3D shape retrieval task, and a new state-of-the-art for the point cloud to surface reconstruction task.

## 1 Introduction

We present a unifying and novel geometry representation for utilizing Convolutional Neural Networks (CNNs) on geometries represented on weighted simplex meshes (including textured point clouds, line meshes, polygonal meshes, and tetrahedral meshes) which preserve maximal shape information based on the Fourier transformation. Most methods that leverage CNNs for shape learning preprocess these shapes into uniform-grid based 2D images (rendered multiview images) or 3D images (binary voxel or Signed Distance Function (SDF)). However, rendered 2D images do not preserve the 3D topologies of the original shapes due to occlusions and the loss of the third spatial dimension. Binary voxels and SDF representations under low resolution suffer big aliasing errors and under high resolution become memory inefficient. Loss of information in the input bottlenecks the effectiveness of the downstream learning process. Moreover, it is not clear how a weighted mesh where each element is weighted by a different scalar or vector (i.e., texture) can be represented by binary voxels and SDF. Mesh and graph based CNNs perform learning on the manifold physical space or graph spectrum, but generality across topologies remains challenging.

In contrast to methods that operate on uniform sampling based representations such as voxel-based and view-based models, which suffer significant representational errors, we use analytical integration to precisely sample in the spectral domain to avoid sample aliasing errors. Unlike graph spectrum based

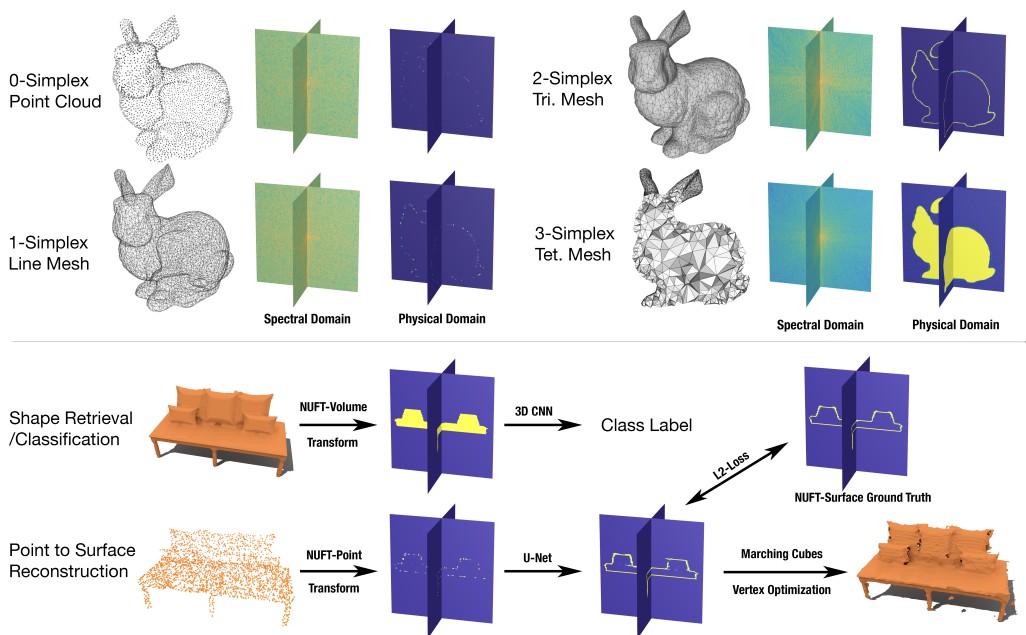

Figure 1: Top: Schematic of the NUFT transformations of the Stanford Bunny model. Bottom: Schematic for shape retrieval and surface reconstruction experiments.

methods, our method naturally generalize across input data structures of varied topologies. Using our representation, CNNs can be directly applied in the corresponding physical domain obtainable by inverse Fast Fourier Transform (FFT) due to the equivalence of the spectral and physical domains. This allows for the use of powerful uniform Cartesian grid based CNN backbone architectures (such as DLA (Yu et al., 2018), ResNet (He et al., 2016)) for the learning task on arbitrary geometrical signals. Although the signal is defined on a simplex mesh, it is treated as a signal in the Euclidean space instead of on a graph, differentiating our framework from graph-based spectral learning techniques which have significant difficulties generalizing across topologies and unable to utilize state-of-the-art Cartesian CNNs.

We evaluate the effectiveness of our shape representation for deep learning tasks with three experiments: a controlled MNIST toy example, the 3D shape retrieval task, and a more challenging 3D point cloud to surface reconstruction task. In a series of evaluations on different tasks, we show the unique advantages of this representation, and good potential for its application in a wider range of shape learning problems. We achieve state-of-the-art performance among non-pre-trained models for the shape retrieval task, and beat state-of-the-art models for the surface reconstruction task.

The key contributions of our work are as follows:

- We develop mathematical formulations for performing Fourier Transforms of signals defined on a simplex mesh, which generalizes and extends to all geometries in all dimensions. (Sec. 3)

- We analytically show that our approach computes the frequency domain representation precisely, leading to much lower overall representational errors. (Sec. 3)

- We empirically show that our representation preserves maximal shape information compared to commonly used binary voxel and SDF representations. (Sec. 4.1)

- We show that deep learning models using CNNs in conjunction with our shape representation achieves state-of-the-art performance across a range of shape-learning tasks including shape retrieval (Sec. 4.2) and point to surface reconstruction (Sec. 4.3)

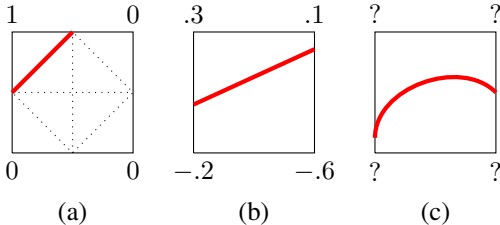

Figure 2: Surface localization in (a) binary pixel/voxel representation, where the boundary can only be in one of $2^4$ (2D) or $2^8$ (3D) discrete locations (b) Signed Distance Function representation, where boundary is linear (c) proposed representation, with nonlinear localization of boundary, achieving subgrid accuracy

| Notation | Description |
|---|---|
| $d$ | Dimension of Euclidean space $\mathbb{R}^d$ |
| $j$ | Degree of simplex. Point $j = 0$, Line $j = 1$, Tri. $j = 2$, Tet. $j = 3$ |
| $n, N$ | Index of the $n$-th element among a total of $N$ elements |
| $\Omega_n^j$ | Domain of $n$-th element of order $j$ |
| $\boldsymbol{x}$ | Cartesian space coordinate vector. $\boldsymbol{x} = (x, y, z)$ |
| $\boldsymbol{k}$ | Spectral domain coordinate vector. $\boldsymbol{k} = (u, v, w)$ |
| $i$ | Imaginary number unit |

Table 1: Notation list

## 2 RELATED WORK

### 2.1 SHAPE REPRESENTATION

Shape learning involves the learning of a mapping from input geometrical signals to desired output quantities. The representation of geometrical signals is key to the learning process, since on the one hand the representation determines the learning architectures, and, on the other hand, the richness of information preserved by the representation acts as a bottleneck to the downstream learning process. While data representation has not been an open issue for 2D image learning, it is far from being agreed upon in the existing literature for 3D shape learning. The varied shape representations used in 3D machine learning are generally classified as multiview images (Su et al., 2015a; Shi et al., 2015; Kar et al., 2017), volumetric voxels (Wu et al., 2015; Maturana & Scherer, 2015; Wu et al., 2016; Brock et al., 2016), point clouds (Qi et al., 2017a;b; Wang et al., 2018b), polygonal meshes (Kato et al., 2018; Wang et al., 2018a; Monti et al., 2017; Maron et al., 2017), shape primitives (Zou et al., 2017; Li et al., 2017), and hybrid representations (Dai & Nießner, 2018).

Our proposed representation is closest to volumetric voxel representation, since the inverse Fourier Transform of the spectral signal in physical domain is a uniform grid implicit representation of the shape. However, binary voxel representation suffers from significant aliasing errors during the uniform sampling step in the Cartesian space (Pantaleoni, 2011). Using boolean values for de facto floating point numbers during CNN training is a waste of information processing power. Also, the primitive-in-cell test for binarization requires arbitrary grouping in cases such as having multiple points or planes in the same cell (Thrun, 2003). Signed Distance Function (SDF) or Truncated Signed Distance Function (TSDF) (Liu et al., 2017; Canelhas, 2017) provides localization for the shape boundary, but is still constrained to linear surface localization due to the linear interpolation process for recovering surfaces from grids. Our proposed representation under Fourier basis can find nonlinear surface boundaries, achieving subgrid-scale accuracy (See Figure 2).

### 2.2 LEARNING ARCHITECTURES

**Cartesian CNNs** are the most ubiquitous and mature type of learning architecture in Computer Vision. It has been thoroughly studied in a range of problems, including image recognition (Krizhevsky et al., 2012; Simonyan & Zisserman, 2014; He et al., 2016), object detection (Girshick, 2015; Ren et al., 2015), and image segmentation (Long et al., 2015; He et al., 2017). In the spirit of 2D image-based deep learning, Cartesian CNNs have been widely used in shape learning models that adopted multiview shape representation (Su et al., 2015a; Shi et al., 2015; Kar et al., 2017; Su et al., 2015b; Pavlakos et al., 2017; Tulsiani et al., 2018). Also, due to its straightforward and analogous extension to 3D by swapping 2D convolutional kernels with 3D counterparts, Cartesian CNNs have also been widely adopted in shape learning models using volumetric representations (Wu et al., 2015; Maturana & Scherer, 2015; Wu et al., 2016; Brock et al., 2016). However, the dense nature of the operations makes it inefficient for sparse 3D shape signals. To this end, improvements to Cartesian

CNNs have been made using space partitioning tree structures, such as Quadtree in 2D and Octree in 3D (Wang et al., 2017; Häne et al., 2017; Tatarchenko et al., 2017). These Cartesian CNNs can leverage backbone CNN architectures being developed in related computer vision problems and thus achieve good performance. Since the physical domain representation in this study is based on Cartesian uniform grids, we directly use Cartesian CNNs.

**Graph CNNs** utilize input graph structure for performing graph convolutions. They have been developed to engage with general graph structured data (Bruna et al., 2013; Henaff et al., 2015; Defferrard et al., 2016). Yi et al. (2017) used spectral CNNs with the eigenfunctions of the graph Laplacian as a basis. However, the generality of this approach across topologies and geometris is still challenging since consistency in eigenfunction basis is implied.

**Specially Designed Neural Networks** have been used to perform learning on unconventional data structures. For example, Qi et al. (2017a) designed a Neural Network architecture for points that achieves invariances using global pooling, with follow-up work (Qi et al., 2017b) using CNNs-inspired hiearchical structures for more efficient learning. Masci et al. (2015) performed convolution directly on the shape manifold and Cohen et al. (2017) designed CNNs for the spherical domain and used it for 3D shapes by projecting the shapes onto the bounding sphere.

## 2.3 FOURIER TRANSFORM OF SHAPE FUNCTIONS

The original work on analytical expressions for Fourier transforms of 2D polygonal shape functions is given by Lee & Mittra (1983). Improved and simpler calculation methods have been suggested in Chu & Huang (1989). A 3D formulation is proposed by Zhang & Chen (2001). Theoretical analyses have been performed for the Fourier analysis of simplex domains (Sun, 2006; Li & Xu, 2009) and Sammis & Strain (2009) designed approximation methods for Fast Fourier Transform of polynomial functions defined on simplices. Prisacariu & Reid (2011) describe shape with elliptic Fourier descriptors for level set-based segmentation and tracking. There has also been a substantial literature on fast non-uniform Fourier transform methods for discretely sampled signal (Greengard & Lee, 2004). However we are the first to provide a simple general expression for a $j$-simplex mesh, an algorithm to perform the transformation, and illustrations of their applicability to deep learning problems.

## 3 REPRESENTATION OF SHAPE FUNCTIONS

### 3.1 MATHEMATICAL PRELIMINARIES

Almost all discrete geometric signals can be abstracted into weighted simplicial complexes. A simplicial complex is a set composed of points, line segments, triangles, and their $d$-dimensional counterparts. We call a simplicial complex consisting solely of $j$-simplices as a homogeneous simplicial $j$-complex, or a $j$-simplex mesh. Most popular geometric representations that the research community is familiar with are simplex meshes. For example, the point cloud is a $0$-simplex mesh, the triangular mesh is a $2$-simplex mesh, and the tetrahedral mesh is a $3$-simplex mesh. A $j$-simplex mesh consists of a set of individual elements, each being a $j$-simplex. If signal is non-uniformly distributed over the simplex, we can define a piecewise constant $j$-simplex function over the $j$-simplex mesh. We call this a weighted simplex mesh. Each element has a distinct signal density.

**J-simplex function** For the $n$-th $j$-simplex with domain $\Omega_n^j$, we define a density function $f_n^j(\boldsymbol{x})$. For example, for some Computer Vision and Graphics applications, a three-component density value can be defined on each element of a triangular mesh for its RGB color content. For scientific applications, signal density can be viewed as mass or charge or other physical quantity.

$$f_n^j(\boldsymbol{x}) = \begin{cases} \rho_n, \boldsymbol{x} \in \Omega_n^j \\ 0, \boldsymbol{x} \notin \Omega_n^j \end{cases} \quad , \quad f^j(\boldsymbol{x}) = \sum_{n=1}^{N} f_n^j(\boldsymbol{x}) \tag{1}$$

The piecewise-constant $j$-simplex function consisting of $N$ simplices is therefore defined as the superposition of the element-wise simplex function. Using the linearity of the integral in the Fourier transform, we can decompose the Fourier transform of the density function on the $j$-simplex mesh to

be a weighted sum of the Fourier transform on individual $j$-simplices.

$$F^j(\boldsymbol{k}) = \int \cdots \int_{-\infty}^{\infty} f^j(\boldsymbol{x}) e^{-i\boldsymbol{k}\cdot\boldsymbol{x}} d\boldsymbol{x} = \sum_{n=1}^{N} \rho_n \underbrace{\int \cdots \int_{\Omega_n^j} e^{-i\boldsymbol{k}\cdot\boldsymbol{x}} d\boldsymbol{x}}_{:=F_n^j(\boldsymbol{k})} = \sum_{n=1}^{N} \rho_n F_n^j(\boldsymbol{k}) \quad (2)$$

## 3.2 Simplex Mesh Fourier Transform

We present a general formula for performing the Fourier transform of signal over a single $j$-simplex. We provide detailed derivation and proof for $j = 0, 1, 2, 3$ in the supplemental material.

$$F_n^j(\boldsymbol{k}) = i^j \gamma_n^j \sum_{t=1}^{j+1} \frac{e^{-i\sigma_t}}{\prod_{l=1,l\neq t}^{j+1}(\sigma_t - \sigma_l)}, \quad \sigma_t := \boldsymbol{k} \cdot \boldsymbol{x}_t \quad (3)$$

We define $\gamma_n^j$ to be the content distortion factor, which is the ratio of content between the simplex over the domain $\Omega_n^j$ and the unit orthogonal $j$-simplex. Content is the $j$-dimensional analogy of the 3-dimensional volume. The unit orthogonal $j$-simplex is defined as a $j$-simplex with one vertex at the Cartesian *origin* and all edges adjacent to the *origin* vertex to be pairwise orthogonal and to have unit length. Therefore from Equation (2) the final general expression for computing the Fourier transform of a signal defined on a weighted simplex mesh is:

$$F^j(\boldsymbol{k}) = \sum_{n}^{N} \rho_n i^j \gamma_n^j \sum_{t=1}^{j+1} \frac{e^{-i\sigma_t}}{\prod_{l=1,l\neq t}^{j+1}(\sigma_t - \sigma_l)} \quad (4)$$

For computing the simplex content, we use the Cayley-Menger Determinant for a general expression:

$$C_n^j = \sqrt{\frac{(-1)^{j+1}}{2^j(j!)^2} \det(\hat{B}_n^j)}, \quad \text{where } \hat{B}_n^j = \begin{bmatrix} 0 & 1 & 1 & 1 & \cdots \\ 1 & 0 & d_{12}^2 & d_{13}^2 & \cdots \\ 1 & d_{21}^2 & 0 & d_{23}^2 & \cdots \\ 1 & d_{31}^2 & d_{32}^2 & 0 & \cdots \\ \vdots & \vdots & \vdots & \vdots & \end{bmatrix} \quad (5)$$

For the matrix $\hat{B}_n^j$, each entry $d_{mn}^2$ represents the squared distance between nodes $m$ and $n$. The matrix is of size $(j+2) \times (j+2)$ and is symmetrical. Since the unit orthogonal simplex has content of $C_I^j = 1/j!$, the content distortion factor $\gamma_n^j$ can be calculated by:

$$\gamma_n^j = \frac{C_n^j}{C_I^j} = j!C_n^j = j!\sqrt{\frac{(-1)^{j+1}}{2^j(j!)^2} \det(\hat{B}_n^j)} \quad (6)$$

**Auxiliary Node Method**: Equation (3) provides a mean of computing the Fourier transform of a simplex with uniform signal density. However, how do we compute the Fourier transform of polytopes (i.e., polygons in 2D, polyhedra in 3D) with uniform signal density efficiently? Here, we introduce the auxiliary node method (AuxNode) that utilizes signed content for efficient computing. We show that for a solid $j$-polytope represented by a watertight $(j-1)$-simplex mesh, we can compute the Fourier transform of the entire polytope by traversing each of the elements in its boundary $(j-1)$-simplex mesh exactly once (Zhang & Chen, 2001).

The auxiliary node method performs Fourier transform over the the signed content bounded by an auxilliary node (a convenient choice being the *origin* of the Cartesian coordinate system) and each $(j-1)$-simplex on the boundary mesh. This forms an auxiliary $j$-simplex: $\Omega_{n'}^j, n' \in [0, N']$, where $N'$ is the number of $(j-1)$-simplices in the boundary mesh. However due to the overlapping of these auxiliary $j$-simplices, we need a means of computing the sign of the transform for the overlapping regions to cancel out. Equation (3) provides a general expression for computing the unsigned transform for a single $j$-simplex. It is trivial to show that since the ordering of the nodes does not affect the determinant in Equation (5), it gives the unsigned content value.

Therefore, to compute the Fourier transform of uniform signals in $j$-polytopes represented by its watertight $(j-1)$-simplex mesh using the auxiliary node method, we modify Equation (3):

$$F_n^j(\boldsymbol{k}) = i^j \sum_{n'=1}^{N_n'} s_{n'} \gamma_{n'}^j \left( \frac{(-1)^j}{\prod_{l=1}^{j} \sigma_l} + \sum_{t=1}^{j} \frac{e^{-i\sigma_t}}{\sigma_t \prod_{l=1,l\neq t}^{j}(\sigma_t - \sigma_l)} \right) \quad (7)$$

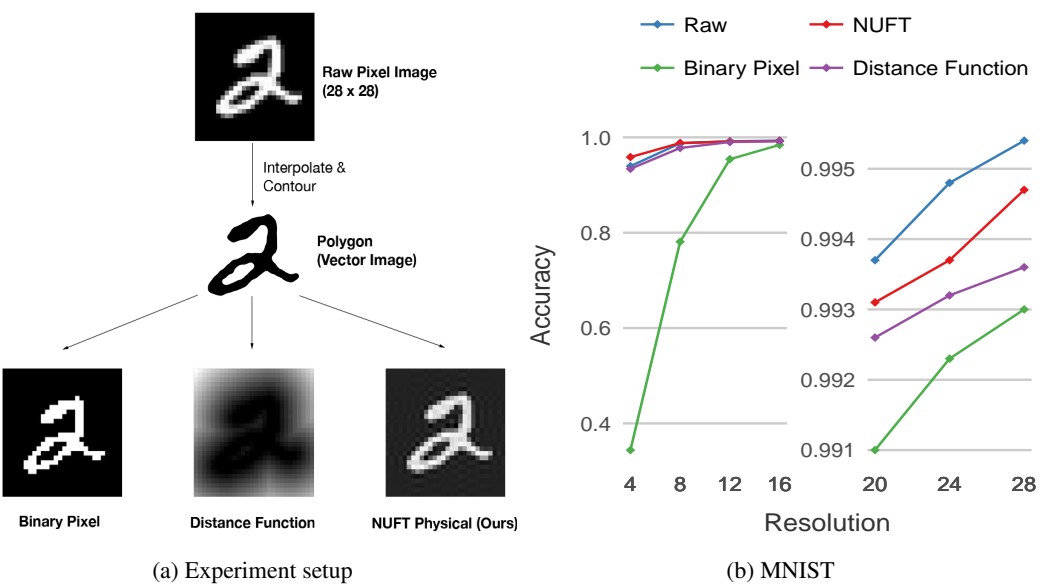

(a) Experiment setup

(b) MNIST

Figure 3: MNIST experiment. (a) schematic for experiment setup. The original MNIST pixel image is up-sampled using interpolation and contoured to get a polygonal representation of the digit. For the polygon, it is transformed into binary pixels, distance functions, and the NUFT physical domain. (b) classification accuracy versus input resolution under various representation schemes. NUFT representation is more optimal, irrespective of resolution.

$s_{n'}\gamma_{n'}^j$ is the signed content distortion factor for the $n'$th auxiliary $j$-simplex where $s_{n'} \in \{-1, 1\}$. For practical purposes, assume that the auxiliary $j$-simplex is in $\mathbb{R}^d$ where $d = j$. We can compute the signed content distortion factor using the determinant of the Jacobian matrix for parameterizing the auxiliary simplex to a unit orthogonal simplex:

$$s_{n'}\gamma_{n'}^j = j! \det(J) = j! \det([\boldsymbol{x}_1, \boldsymbol{x}_2, \cdots, \boldsymbol{x}_j]) \tag{8}$$

Since this method requires the boundary simplices to be oriented, the right-hand rule can be used to infer the correct orientation of the boundary element. For 2D polygons, it requires that the watertight boundary line mesh be oriented in a counter-clockwise fashion. For 3D polytopes, it requires that the face normals of boundary triangles resulting from the right-hand rule be consistently outward-facing.

**Algorithmic implementation:** Several efficiencies can be exploited to achieve fast runtime and high robustness. First, since the general expression Equation (4) involves division and is vulnerable to division-by-zero errors (that is not a singularity since it can be eliminated by taking the limit), add a minor random noise $\epsilon$ to vertex coordinates as well as to the $\boldsymbol{k} = \boldsymbol{0}$ frequency mode for robustness. Second, to avoid repeated computation, the value should be cached in memory and reused, but caching all $\sigma$ and $e^{-i\sigma}$ values for all nodes and frequencies is infeasible for large mesh and/or high resolution output, thus the Breadth-First-Search (BFS) algorithm should be used to traverse the vertices for efficient memory management.

## 4 EXPERIMENTS

In this section, we will discuss the experiment setup, and we defer the details of our model architecture and training process to the supplementary material since it is not the focus of this paper.

### 4.1 MNIST WITH POLYGONS

We use the MNIST experiment as a first example to show that shape information in the input significantly affects the efficacy of the downstream learning process. Since the scope of this research is on efficiently learning from nonuniform mesh-based representations, we compare our method with the state of the art in a slightly different scenario by treating MNIST characters as polygons. We

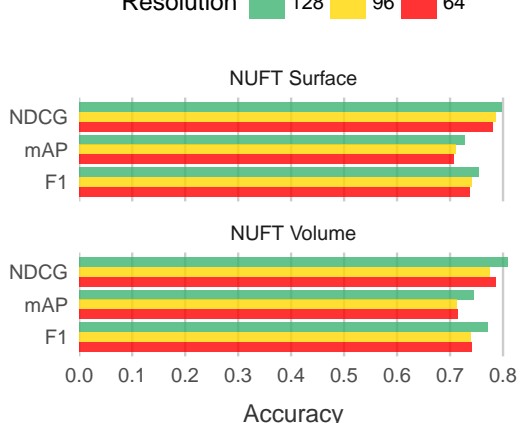

| Rep | Method | F1 | mAP | NDCG |
|---|---|---|---|---|
| | *No Pre-training* | | | |
| Volu- | Ours(NUFT-V) | **0.770** | **0.745** | **0.809** |
| metric | DeepVoxNet | 0.253 | 0.192 | 0.277 |
| | *With Pre-training* | | | |
| | RotationNet | **0.798** | **0.772** | **0.865** |
| Multi- | ImprovGIF | 0.767 | 0.722 | 0.827 |
| View | ReVGG | 0.772 | 0.749 | 0.828 |
| | MVCNN | 0.764 | 0.735 | 0.815 |

Figure 4: Comparison between NUFT Volume and NUFT Surface performance at different resolutions.

Table 2: Comparison of shape retrieval performance with state-of-the-art models. The best result among each representation category is highlighted in bold.

choose this experiment as a first toy example since it is easy to control the learning architecture and input resolution to highlight the effects of shape representation on deep learning performance.

**Experiment setup**: We pre-process the original MNIST raw pixel images into polygons, which are represented by watertight line meshes on their boundaries. The polygonized digits are converted into $(n \times n)$ binary pixel images and into distance functions by uniformly sampling the polygon at the $(n \times n)$ sample locations. For NUFT, we first compute the lowest $(n \times n)$ Fourier modes for the polygonal shape function and then use an inverse Fourier transform to acquire the physical domain image. We also compare the results with the raw pixel image downsampled to different resolutions, which serves as an oracle for information ceiling. Then we perform the standard MNIST classification experiment on these representations with varying resolution $n$ and with the same network architecture.

**Results**: The experiment results are presented in Figure (3b). It is evident that binary pixel representation suffers the most information loss, especially at low resolutions, which leads to rapidly declining performance. Using the distance function representation preserves more information, but underperforms our NUFT representation. Due to its efficient information compression in the Spectral domain, NUFT even outperforms the downsampled raw pixel image at low resolutions.

## 4.2 3D SHAPE RETRIEVAL

Shape retrieval is a classic task in 3D shape learning. SHREC17 (Savva et al., 2016) which is based on the ShapeNet55 Core dataset serves as a compelling benchmark for 3D shape retrieval performance. We compare the retrieval performance of our model utilizing the NUFT-surface (NUFT-S) and NUFT-volume (NUFT-V) at various resolutions against state-of-the-art shape retrieval algorithms to illustrate its potential in a range of 3D learning problems. We performed the experiments on the normalized dataset in SHREC17. Our model utilizes 3D DLA (Yu et al., 2018) as a backbone architecture.

**Results:** Results from the experiment are tabulated in Table 2. For the shape retrieval task, most state-of-the-art methods are based on multi-view representation that utilize a 2D CNN pretrained on additional 2D datasets such as ImageNet. We have achieved results on par with, though not better than, state-of-the-art pretrained 2D models. We outperform other models in this benchmark that have not been pre-trained on additional data. We also compared NUFT-volume and NUFT-surface representations in Figure 4. Interestingly NUFT-volume and NUFT-surface representations lead to similar performances under the same resolution.

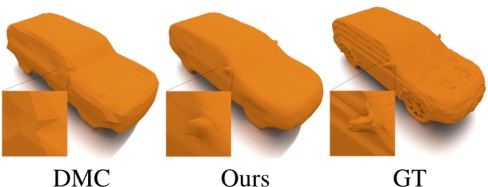

Figure 5: Zoom-in comparison.

| Method | Chamfer | Accuracy | Complete |
|---|---|---|---|
| DMC | 0.218 | 0.182 | 0.254 |
| PSR-5 | 0.352 | 0.405 | 0.298 |
| PSR-8 | 0.198 | 0.196 | 0.200 |
| Ours(w/ Noise) | **0.144** | 0.150 | **0.137** |
| Ours(w/o Noise) | 0.145 | **0.125** | 0.165 |

Table 3: Quantitative comparison of surface reconstruction methods. The metrics above are distances, hence lower value represents better performance. Best result is highlighted in bold. We achieve better results than the current state-of-the-art method by a sizable margin, and our results are robust to noise in the input.

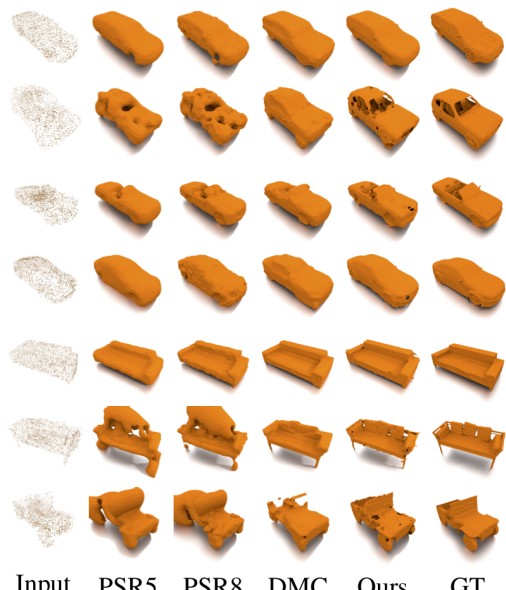

Input    PSR5    PSR8    DMC    Ours    GT

Figure 6: Qualitative side-by-side comparison of surface reconstruction results.

## 4.3 3D Surface Reconstruction from Point Clouds

We further illustrate the advantages of our representation with a unique yet important task in computational geometry that has been challenging to address with conventional deep learning techniques: surface reconstruction from point cloud. The task is challenging for deep learning in two aspects: First, it requires input and output signals of different topologies and structures (i.e., input being a point cloud and output being a surface). Second, it requires precise localization of the signal in 3D space. Using our NUFT-point representation as input and NUFT-surface representation as output, we can frame the task as a 3D image-to-image translation problem, which is easy to address using analogous 2D techniques in 3D. We use the U-Net (Ronneberger et al., 2015) architecture and train it with a single $L_2$ loss between output and ground truth NUFT-surface representation.

**Experiment Setup**: We train and test our model using shapes from three categories of ShapeNet55 Core dataset (car, sofa, bottle). We trained our model individually for these three categories. As a pre-processing step we removed faces not visible from the exterior and simplified the mesh for faster conversion to NUFT representation. For the input point cloud, we performed uniform point sampling of 3000 points from each mesh and converted the points into the NUFT-point representation ($128^3$). Then, we converted the triangular mesh into NUFT-surface representation ($128^3$). At test time, we post-process the output NUFT-surface implicit function by using the marching cubes algorithm to extract $0.5$ contours. Since the extracted mesh has thickness, we further shrink the mesh by moving vertices to positions with higher density values while preserving the rigidity of each face. Last but not least, we qualitatively and quantitatively compare the performance by showing the reconstructed mesh against results from the traditional Poisson Surface Reconstruction (PSR) method (Kazhdan & Hoppe, 2013) at various tree depths (5 and 8) and the Deep Marching Cubes (DMC) algorithm (Liao et al., 2018). For quantitative comparison, we follow the literature (Seitz et al., 2006) and use Chamfer distance, Accuracy and Completeness as the metrics for comparison. For comparison with Liao et al. (2018), we also test the model with noisy inputs (Gaussian of sigma 0.15 voxel-length under 32 resolution), computed distance metrics after normalizing the models to the range of (0, 32).

**Results**: Refer to Table 3 for quantitative comparisons with competing algorithms on the same task, and Figures 5 and 6 for visual comparisons. GT stands for Ground Truth. We achieve new state-of-the-art in the point to surface reconstruction task, due to the good localization properties of the NUFT representations and its flexibility across geometry topologies.

## 5 CONCLUSION

We present a general representation for multidimensional signals defined on simplicial complexes that is versatile across geometrical deep learning tasks and maximizes the preservation of shape information. We develop a set of mathematical formulations and algorithmic tools to perform the transformations efficiently. Last but not least, we illustrate the effectiveness of the NUFT representation with a well-controlled example (MNIST polygon), a classic 3D task (shape retrieval) and a difficult and mostly unexplored task by deep learning (point to surface reconstruction), achieving new state-of-the-art performance in the last task. In conclusion, we offer an alternative representation for performing CNN based learning on geometrical signals that shows great potential in various 3D tasks, especially tasks involving mixed-topology signals.

## ACKNOWLEDGEMENTS

We would like to thank Yiyi Liao for helping with the DMC comparison, Jonathan Shewchuk for valuable discussions, and Luna Huang for LATEXmagic. Chiyu "Max" Jiang is supported by the Chang-Lin Tien Graduate Fellowship and the Graduate Division Block Grant Award of UC Berkeley. This work is supported by a TUM-IAS Rudolf Mößbauer Fellowship and the ERC Starting Grant *Scan2CAD* (804724).

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

# APPENDIX

## A  MATHEMATICAL DERIVATION

Without loss of generality assume that the $j$-simplex is defined in $\mathbb{R}^d$ space where $d \geq j$, since it is not possible to define a $j$-simplex in a space with dimensions lower than $j$. For most cases below (except $j = 0$) we will parameterize the original simplex domain to a unit orthogonal simplex in $\mathbb{R}^j$ (as shown in Figure 7). Denote the original coordinate system in $\mathbb{R}^d$ as $\boldsymbol{x}$ and the new coordinate system in the parametric $\mathbb{R}^j$ space as $\boldsymbol{p}$. Choose the following parameterization scheme:

$$\boldsymbol{x}(\boldsymbol{p}) = [\boldsymbol{x}_1 - \boldsymbol{x}_d, \boldsymbol{x}_2 - \boldsymbol{x}_d, \cdots, \boldsymbol{x}_{d-1} - \boldsymbol{x}_d]\boldsymbol{p} + \boldsymbol{x}_d \tag{9}$$

By performing the Fourier transform integral in the parametric space and restoring the results by the content distortion factor $\gamma_n^j$ we can get equivalent results as the Fourier transform on the original simplex domain. Content is the generalization of volumes in arbitrary dimensions (i.e. unity for points, length for lines, area for triangles, volume for tetrahedron). Content distortion factor $\gamma_n^j$ is the ratio between the content of the original simplex and the content of the unit orthogonal simplex in parametric space. The content is signed if switching any pair of nodes in the simplex changes the sign of the content, and it is unsigned otherwise. See subsection 3.2 for means of computing the content and the content distortion factor.

### A.1  POINT: $0$-SIMPLEX

Points have spatial position $\boldsymbol{x}$ but no size (length, area, volume), hence it can be mathematically modelled as a delta function. The delta function (or Dirac delta function) has point mass as a function equal to zero everywhere except for zero and its integral over entire real space is one, i.e.:

$$\iiint_{-\infty}^{\infty} \delta(\boldsymbol{x})d\boldsymbol{x} = \iiint_{0-}^{0+} \delta(\boldsymbol{x})d\boldsymbol{x} = 1 \tag{10}$$

For unit point mass at location $\boldsymbol{x}_n$:

$$f_n^0(\boldsymbol{x}_n) = \delta(\boldsymbol{x} - \boldsymbol{x}_n) \tag{11}$$

$$F_n^0(\boldsymbol{\omega}) = \iiint_{-\infty}^{\infty} f_n^0(\boldsymbol{x})e^{-i\boldsymbol{\omega}\cdot\boldsymbol{x}}d\boldsymbol{x} \tag{12}$$

$$= \iiint_{-\infty}^{\infty} \delta(\boldsymbol{x} - \boldsymbol{x}_n)e^{-i\boldsymbol{\omega}\cdot\boldsymbol{x}}d\boldsymbol{x} \tag{13}$$

$$= e^{-i\boldsymbol{\omega}\cdot\boldsymbol{x}_n} \tag{14}$$

Indeed, for $0$-simplex, we have recovered the definition of the Discrete Fourier Transform (DFT).

### A.2  LINE: $1$-SIMPLEX

For a line with vertices at location $\boldsymbol{x}_1, \boldsymbol{x}_2 \in \mathbb{R}^d$, by parameterizing it onto a unit line, we get:

$$F_n^1(\boldsymbol{\omega}) = \gamma_n^1 \int_0^1 dp \, e^{-i\boldsymbol{\omega}\cdot\boldsymbol{x}(\boldsymbol{p})} \tag{15}$$

$$= i\gamma_n^1 \left( \frac{e^{-i\boldsymbol{\omega}\cdot\boldsymbol{x}_1}}{\omega(\boldsymbol{x}_1 - \boldsymbol{x}_2)} + \frac{e^{-i\boldsymbol{\omega}\cdot\boldsymbol{x}_2}}{\omega(\boldsymbol{x}_2 - \boldsymbol{x}_1)} \right) \tag{16}$$

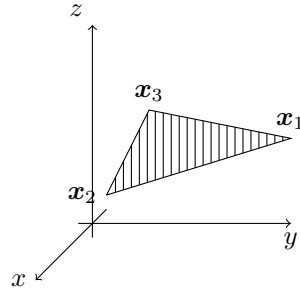 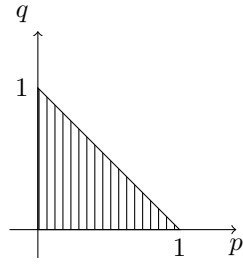

(a) Original $j$-simplex in $\mathbb{R}^d$ space, $d \geq j$.  (b) Unit orthogonal $j$-simplex in $\mathbb{R}^j$

Figure 7: Schematic of example for 2-simplex. Original $j$-simplex is parameterized to a unit orthogonal $j$-simplex in $\mathbb{R}^j$ space for performing the integration. Parameterization incurs a content distortion factor $\gamma_n^j$ which is the ratio between the original simplex and the unit-orthogonal simplex in parametric space.

### A.3 TRIANGLE: 2-SIMPLEX

For a triangle with vertices $\boldsymbol{x}_1, \boldsymbol{x}_2, \boldsymbol{x}_3 \in \mathbb{R}^d$, parameterization onto a unit orthogonal triangle gives:

$$F_n^2(\boldsymbol{\omega}) = \gamma_n^2 \int_0^1 dp \int_0^{1-p} dq \, e^{-i\boldsymbol{\omega}\cdot\boldsymbol{x}(\boldsymbol{p})} \tag{17}$$

$$= \gamma_n^2 e^{-i\boldsymbol{\omega}\cdot\boldsymbol{x}_3} \int_0^1 dp \, e^{-ip\boldsymbol{\omega}\cdot(\boldsymbol{x}_1-\boldsymbol{x}_3)} \int_0^{1-p} dq \, e^{-iq\boldsymbol{\omega}\cdot(\boldsymbol{x}_2-\boldsymbol{x}_3)} \tag{18}$$

$$= -\gamma_n^2 \left( \frac{e^{-i\boldsymbol{\omega}\cdot\boldsymbol{x}_1}}{\boldsymbol{\omega}^2(\boldsymbol{x}_1-\boldsymbol{x}_2)(\boldsymbol{x}_1-\boldsymbol{x}_3)} + \frac{e^{-i\boldsymbol{\omega}\cdot\boldsymbol{x}_2}}{\boldsymbol{\omega}^2(\boldsymbol{x}_2-\boldsymbol{x}_1)(\boldsymbol{x}_2-\boldsymbol{x}_3)} + \frac{e^{-i\boldsymbol{\omega}\cdot\boldsymbol{x}_3}}{\boldsymbol{\omega}^2(\boldsymbol{x}_3-\boldsymbol{x}_1)(\boldsymbol{x}_3-\boldsymbol{x}_2)} \right) \tag{19}$$

### A.4 TETRAHEDRON: 3-SIMPLEX

For a tetrahedron with vertices $\boldsymbol{x}_1, \boldsymbol{x}_2, \boldsymbol{x}_3, \boldsymbol{x}_4 \in \mathbb{R}^d$, parameterization onto a unit orthogonal tetrahedron gives:

$$F_n^3(\boldsymbol{\omega}) = \gamma_n^3 \int_0^1 dp \int_0^{1-p} dq \int_0^{1-p-q} dr \, e^{-i\boldsymbol{\omega}\cdot\boldsymbol{x}(\boldsymbol{p})} \tag{20}$$

$$= \gamma_n^3 e^{-i\boldsymbol{\omega}\cdot\boldsymbol{x}_4} \int_0^1 dp \, e^{-ip\boldsymbol{\omega}\cdot(\boldsymbol{x}_1-\boldsymbol{x}_4)} \int_0^{1-p} dq \, e^{-iq\boldsymbol{\omega}\cdot(\boldsymbol{x}_2-\boldsymbol{x}_4)} \int_0^{1-p-q} dr e^{-ir\boldsymbol{\omega}\cdot(\boldsymbol{x}_3-\boldsymbol{x}_4)} \tag{21}$$

$$= \frac{-i\gamma_n^3 \left( \dfrac{e^{-i\boldsymbol{\omega}\cdot\boldsymbol{x}_1}}{\boldsymbol{\omega}^3(\boldsymbol{x}_1-\boldsymbol{x}_2)(\boldsymbol{x}_1-\boldsymbol{x}_3)(\boldsymbol{x}_1-\boldsymbol{x}_3)} + \dfrac{e^{-i\boldsymbol{\omega}\cdot\boldsymbol{x}_2}}{\boldsymbol{\omega}^3(\boldsymbol{x}_2-\boldsymbol{x}_1)(\boldsymbol{x}_2-\boldsymbol{x}_3)(\boldsymbol{x}_2-\boldsymbol{x}_4)} +}{\dfrac{e^{-i\boldsymbol{\omega}\cdot\boldsymbol{x}_3}}{\boldsymbol{\omega}^3(\boldsymbol{x}_3-\boldsymbol{x}_1)(\boldsymbol{x}_3-\boldsymbol{x}_2)(\boldsymbol{x}_3-\boldsymbol{x}_4)} + \dfrac{e^{-i\boldsymbol{\omega}\cdot\boldsymbol{x}_4}}{\boldsymbol{\omega}^3(\boldsymbol{x}_4-\boldsymbol{x}_1)(\boldsymbol{x}_4-\boldsymbol{x}_2)(\boldsymbol{x}_4-\boldsymbol{x}_3)} \right)} \tag{22}$$

## B    COMPARISON OF GEOMETRICAL SHAPE INFORMATION

Besides evaluating and comparing the shape representation schemes in the context of machine learning problems, we evaluate the different representation schemes in terms of its geometrical shape information. To evaluate geometrical shape information, we first convert the original polytopal shapes into the corresponding representations and then reconstruct the shape from these representations using interpolation based upsampling followed by contouring methods. Finally, we use the mesh Intersection over Union (mesh-IoU) metric between the original and constructed mesh to quantify geometrical shape information. Mesh boolean and volume computation for 2D polygons and 3D triangular mesh can be efficiently performed with standard computational geometry methods. In three dimensions contouring and mesh extraction can be performed using marching cubes algorithm for mesh reconstruction. For binarized representation, we perform bilinear upsampling (which does not affect the final result) followed by 0.5-contouring. For our NUFT representation, we perform spectral domain upsampling (which corresponds to zero-padding of higher modes in spectral domain), followed by 0.5-contouring. Qualitative side-by-side comparisons are presented for visual inspection, and qualitative empirical evaluation is performed for the Bunny Mesh (1K faces) model. Refer to Figures 8, 9, and 10.

## C    MODEL ARCHITECTURE AND TRAINING DETAILS

### C.1    MNIST WITH POLYGONS

**Model Architecture:**    We use the state-of-the-art Deep Layer Aggregation (DLA) backbone architecture with [1,1,2,1] levels and [32, 64, 256, 512] filter numbers. We keep the architecture constant while varying the input resolution.

**Training Details:**    We train the model with batch size of 64, learning rate of 0.01 and learning rate step size of 10. We use the SGD optimizer with momentum of 0.9, weight decay of $1 \times 10^{-4}$ for 15 epochs.

### C.2    3D SHAPE RETRIEVAL

**Model Architecture:**    We use DLA34 architecture with all 2D convolutions modified to be 3D convolutions. It consists of [1,1,1,2,2,1] levels, with [16, 32, 64, 128, 256, 512] filter numers.

**Training Details**    We train the model with batch size of 64, learning rate of $1 \times 10^{-3}$, learning rate step size of 30. We use the Adam optimizer with momentum of 0.9, and weight decay of $1 \times 10^{-4}$ for 40 epochs.

### C.3    3D SURFACE RECONSTRUCTION FROM POINT CLOUD

**Model Architecture**    We use a modified 3D version of the U-Net architecture consisting of 4 down convolutions and 4 up-convolutions with skip layers. Number of filters for down convolutions are [32, 64, 128, 256] and double of that for up convolutions.

**Training Details**    We train the model using Adam optimizer with learning rate $3 \times 10^{-4}$ for 200 epochs. We use NUFT point representation as input and a single $L_2$ loss between output and ground truth (NUFT surface) to train the network. We train and evaluate the model at $128^3$ resolution.

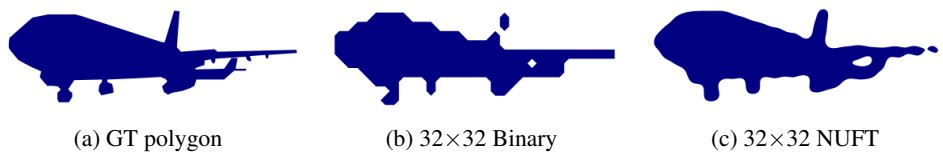

(a) GT polygon        (b) 32×32 Binary        (c) 32×32 NUFT

Figure 8: Visualizing different representations. (a) Shows the original ground truth polygon, (b, c) show reconstructed polygons from binary and NUFT representations.

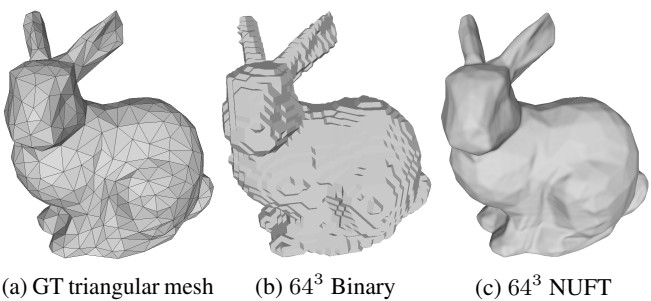

(a) GT triangular mesh    (b) $64^3$ Binary    (c) $64^3$ NUFT

Figure 9: Comparison between 3D shapes. (a) Original mesh, (b) Reconstructed mesh from Binary Voxel ($64 \times 64 \times 64$), (c) Reconstructed mesh from NUFT ($64 \times 64 \times 64$)

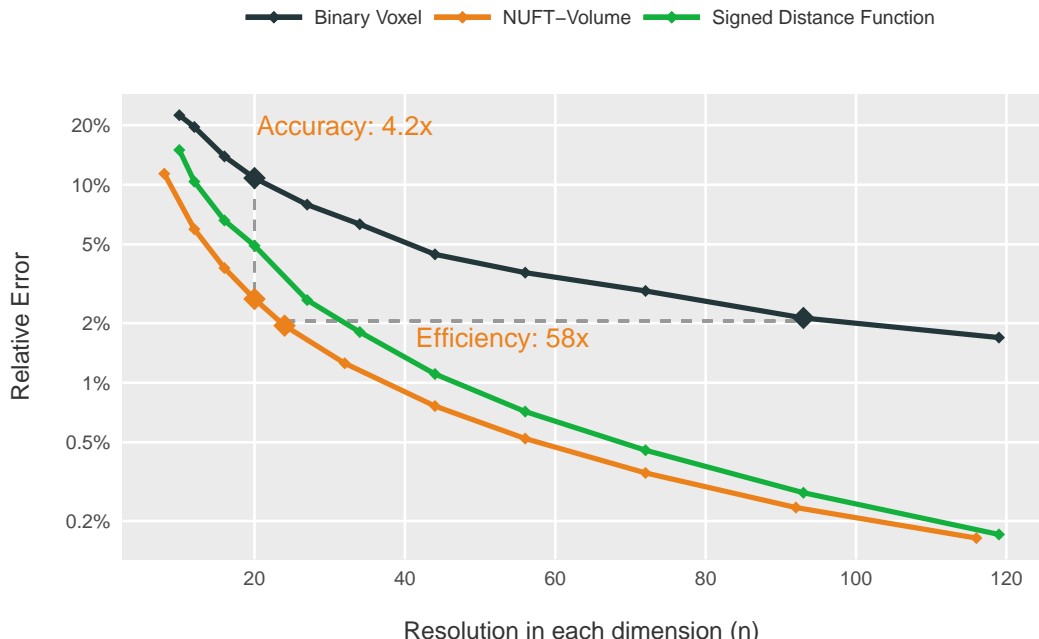

Figure 10: Comparison of Representations in Mesh Recovery Accuracy (Example mesh: Stanford Bunny 1K Mesh). Notes: (i) Relative error is defined by the proportion of volume of differenced mesh to volume of the original mesh. (ii) Error estimates for NUFT-Volume over 50 on abscissa are inaccurate due to inadequate quadrature resolution.

