# OpenReview forum: "Convolutional Neural Networks on Non-uniform Geometrical Signals Using Euclidean Spectral Transformation"
_ICLR.cc/2019/Conference_

### Official Review · AnonReviewer3 · 2018-11-03
**Hard to Understand, insufficient results.**

**Rating:** 4
**Confidence:** 3

**Review:**

This paper introduces a method for handing input data that is defined on irregular mesh-type domains. If I understand it correctly, the core technique is  to perform Fourier analysis to transform the input data to the frequency domain, which is then transformed back to a regular domain before applying a standard neural network. The claimed result is that this is better than standard linear interpolations. The key technical contribution is to define FT on points, edges, and meshes (This reviewer appreciates these efforts). Explicit formula are given. However, the paper does not perform convolutions on the input irregular domain directly, which is quite disappointing. The experimental results are preliminary. It is expected to perform evaluation on more applications such as semantic segmentation.


The major issue of the paper is that the goal was not stated clearly. Does it target for a neural network that is defined on irregular domains or simply a technique to handling irregular domains? Given the Fourier transform, it is possible to define convolutions directly as multiplications in the Fourier domain....the paper can be more interesting, if it follows this line.

Overall, it is hard to champion the paper based on the current technical approach and the experimental results.

---

> ### Author Response · Authors · 2018-11-26
> **Re: Hard to Understand, insufficient results**
>
> We thank you for your review and feedback, and we hope to be able to address your concerns below.
>
> Our paper addresses the issue of handling irregular domains, with possibly mixed topologies in the context of deep learning, and propose an optimal spectral sampling scheme for constructing a volumetric representation in structured grids. Our experiments are chosen to reflect these as aspects. i.e., MNIST experiment compares our representation to conventional schemes (binary pixel, SDF) to show that our representation conserves more shape information and is more robust to limited resolution. Shape retrieval experiment shows the applicability of our method to standard classification tasks, and the surface reconstruction task highlights the ability of our framework to handle mixed topologies. The fact that limits our choice to performing CNNs in the physical domain is that even though the convolution step can be performed in the spectral domain, nonlinearities cannot. Hence in order to perform spectral convolution there needs to be repeated forward and inverse transformations, rendering the process less efficient.

---

### Official Review · AnonReviewer1 · 2018-11-03
**positive solution to learn across varied graph topologies**

**Rating:** 7
**Confidence:** 4

**Review:**

Convolutiuonnal Neural Networks on Non-uniform Geometrical Signals using Euclidean Spectral Transformation

The paper tackles the challenging problem of learning across mixed graph topologies, which is today a real challenge. It is highly original due to the unified general framework for handling differing graph topologies. The method is highly generalizable to other learning techniques since it proposes a transformation of varied topologies into a cartesian grid-like embedding, via the new non-uniform Fourier transform. The method is evaluated on MNIST, Shape Retrieval, and point to surface reconstruction. The paper is dense and uses non-trivial mathematical formulations, but reads well and remains easy to follow. The experiments supports well, and greatly adds clarity to understand the proposed methodology. Overall, recommendation towards acceptance.

Positive
+ Develop a method to analyze signal on mixed topologies with a new non-uniform Fourier transform.
+ The proposed approach has advantages on -reducing sampling error, -unified framework for mixed topology, -reducing heuristics in designing can architecture, -local weights in mesh structures.
+ Improved performance in surface reconstruction task.

Specific Comments
- In the inverse Fourier Transformation, a voxel-like grid structure is used, however - how to control the size of this volume? If the size is large, or explodes, the complexity of the cnn architecture would explode as well - How is this size issue tackled?
- In this same inverse Fourier Transformation, the whole infinite space would be obviously hard to sample - Spectral information would be lost - How bad is this and how does this impact results?  How would this compare to direct graph-based methods, for instance, in a fixed graph structure?
- Ovsjanikov’s Functional maps, siggraph 2012, have been proposed to find maps between differing graph spectrum, partly solving the problem of handling graphs of multiple topologies. One way would be to find spectral correspondences between embeddings. How helpful would this be to find similarities between embeddings in this new proposed unified framework?

Typos - geometris

---

> ### Author Response · Authors · 2018-11-26
> **Re: positive solution to learn across varied graph topologies**
>
> Thanks for your helpful feedback and suggestions! We will try to address your various comments below:
>
> - Size of the grid structure
> The size, or resolution of the given representation, is predetermined prior to running the CNN. Higher resolution leads to improved performance (see Fig. 3, 4) but also higher computational and memory budget. 128^3 resolution can comfortably fit in Modern GPUs (such as NVIDIA 1080 Ti). Also as inverse FFT is concerned, it is an extremely efficiently algorithm (o(nlogn)) and reasonable resolutions (such as 128^3) is manageable.
>
> - Spectra truncation
> We have compared the results of varied resolution in Fig. 3, 4, where higher resolution generally results in higher performance. Also Fig. 3 compares the effect of resolution on different shape representations, where our NUFT representation is less sensitive to resolution. Unfortunately, to our knowledge, graph-based methods for the given tasks are not available in the literature for comparison.
>
> - Functional maps
> Ideas of using functional maps towards deep learning have been explored (SyncSpecCNN, Yi et al. 2017). However, it targets a fundamentally different scenario. Functional maps in this context only concerns different topologies of the surface manifold (i.e., across genuses), whereas the different topologies in our case are the different degrees of simplex units (points, lines, surfaces, volumes), irrespective of shape genus.

---

### Official Review · AnonReviewer2 · 2018-11-06
**This paper concerns the application of standard CNN methods to input from geometric domains where information is not necessarily provided pre-sampled on a Euclidean grid.**

**Rating:** 5
**Confidence:** 3

**Review:**

Overall Thoughts:

I think this paper addresses an interesting topic and that the community as a whole are interested on the application of learning algorithms to non-Euclidean domains. It is nice to see the application of Fourier sampling to geometric primitives in a sensible manner and I am positive about that part of the paper. However, in its current form, I have quite a few questions about the approach and the empirical studies - I would need to here more information from the authors on the points below.

Specific Comments/Questions:

Sec1: The authors make a number of assertions about the representational errors that occur in other approaches - I feel that these claims should be supported by specific references.

Contributions: It is not clear to me that the experiments show that the method “preserves maximal information content” - in my understanding, information content has a formal definition and I don’t see where this is presented in the results?

Sec2: Before CNNs there has been a substantial analysis of Fourier methods applied to shape models, e.g. elliptical Fourier series for shape contours (and signed distance representations) by Prisacariu et al.

Sec2: It is also worth noting that there is substantial literature on non-uniform Fourier methods including the non-uniform Fourier transform and a number of accelerations (e.g. NUFFT) as well as consideration of the implications of band-limiting the sampled spectra.

Sec3: The mathematical derivation all makes sense to me and makes use of results for piecewise uniform signals. Please could the authors provide more details on how the spectra are represented? These discontinuous signals (esp. delta functions) will have infinite bandwidth in the spectral domain so how they are stored would seem very important to me. Are the signals band limited at some point? If so, how does this affect the approximation and should filtering/windowing be used? Otherwise is there not a difficult storage problem? The final paragraph suggests all the analytic signals might be stored but this has a big impact on how efficient the algorithm is and really is far too important a point to just have a single sentence - please can the authors expand on how this is actually implemented and what the computational considerations are (and resulting impacts on performance)?

Sec4: Please can the authors add error bars (at the least) to all the tables/plots in the results. It is entirely unreasonable to make any statements about how significant the results may be without even the most basic of analysis. Ideally we should see histograms of the results for the retrieval and shape reconstruction results.

Sec4: How is the downsampling performed in the MNIST experiment? It would seem very important to take care with this for the purpose of comparison. A significant disparity between the signed distance function and the NUFT would seem slightly surprising to me? Again, without error bars we really cannot say very much about the results on the right of Fig3(b).

Sec4: Could the Fig4 results not be provided as histograms? We also need many more details about how the results were obtained and procedures to ensure that the results are meaningful and robust (e.g. repeated tests and partitions of the data, etc..)

Arch and Training Details: How are we to know that these choices provide fair comparisons to previous approaches?

Sec3/4: All the results seem to require the input to be reconstructed back into a dense sampling domain (via inverse FT) - is this the case? Would it not be more efficient to perform the convolutions in the spectral domain where the signal is sparse?

Sec4: It seems pretty unfair to train and test on a single category of shapes in the shape test since the data is known not to be very diverse? Particularly when, unless I’ve misunderstood, the baselines on the shape recovery test to not involve learning and so (while helpful to have) they are not really fair baselines compared to other learning approaches? Also, please can the authors provide much more information about the extra processing applied (e.g. the part starting with the “extra mesh thickness”) since there seem to be some extra steps that are nothing to do with the rest of the method and may impact the results significantly.

Sec4: It is interesting that Table 3 (again difficult to say without error bars) indicates that there are times when the method performs better with the additional of noise - this seems counter-intuitive - please could the authors comment on this?

Other Points:

I’m afraid that there are quite a few grammatical errors in the text (too many to list here) so I would recommend another round of proof-reading.

---

> ### Author Response · Authors · 2018-11-26
> **Re: This paper concerns the application of standard CNN methods to input from geometric domains where information is not necessarily provided pre-sampled on a Euclidean grid.**
>
> We appreciate your thorough review and helpful suggestions. We will try to address you questions and suggestions below:
> - Sec.1 Representational Error
> Our NUFT transformation of simplex meshes analytically and exactly computes all spectral coefficients under a certain cut-off frequency, hence it is an ideal low-pass filter, resulting in no aliasing error due to the strict containment of frequencies below the cut-off frequency. We acknowledge that the term “information content” could be potentially misleading, and replace it with “shape information” instead. We provide empirical evidence of the effects of such shape information on machine learning tasks (MNIST experiment) as well as purely for shape reconstruction purpose (Appendix B & Figure 10).
>
> -Sec. 2 Additional references
> We appreciate the additional references provided, and added them to the literature overview section (see Sec. 2).
>
> -Sec. 3 Representing the spectra
> The spectra is represented as the spectral coefficients that are limited under a certain cut-off frequency, hence the signal is band-limited. Exactly computing only spectral coefficients under a cut-off frequency followed by inverse transform to the physical domain is equivalent to filtering the original signal (defined on a simplex mesh) with an ideal low-pass filter. We show the effects of truncating the spectrum on deep learning tasks in experiments (MNIST and shape retrieval by varying input resolution).
>
> -Sec. 4 Error bars
> Adding error bars to figures is unfortunately difficult to accomplish as the deep learning algorithms are deterministic in the current form. Models in the vision literature are usually much more deterministic and reproducible than those of other fields such as reinforcement learning. Moreover the baseline models we compared to do not have error bars in the original papers, and it is non-trivial to reproduce their results since most of them did not release source code.
>
> -Sec. 4 Downsampling
> In the paper, we described the sampling process for the various representations: “The polygonized digits are converted into (n × n) binary pixel images and into distance functions by uniformly sampling the polygon at the (n × n) sample locations. For NUFT, we first compute the lowest (n × n) Fourier modes for the polygonal shape function and then use an inverse Fourier transform to acquire the physical domain image.” By downsampling, we mean simply using a smaller value of n in the conversion. The disparity between NUFT and SDF is not too huge, as compared to the disparity between NUFT and Binary Pixels.
>
> -Sec. 4 Fig. 4
> Detailed values for the best run in Fig 4 is provided in Table 2 for comparison with baseline models. Fig 4 provides a comparison between input NUFT representations (volume vs surface) and shows that the difference between the representations are small. We use the official partition of the data for the experiment for fair comparison with baselines.
>
> -Arc and training details
> The DLA backbone architecture is a state-of-the-art CNN architecture that performs well in this task. As fairness is concerned, in the MNIST experiment, we controlled all aspects of the experiment (architecture, training schedule, etc.) except for input representation to study the effects of shape representation, hence the same architecture is used for all cases.
>
> -Sec. 3/4 Reconstructed back into a dense sampling domain
> This is indeed the case. The fact that limits our choice to performing CNNs in the physical domain is that even though the convolution step can be performed in the spectral domain, nonlinearities cannot. Hence in order to perform spectral convolution there needs to be repeated forward and inverse transformations, rendering the process less efficient.
>
> -Sec. 4 Baseline algorithms
> The baseline PSR algorithm is not learning based, and training on single categories has its limitations. But for targeted application scenarios (indoor structure inference etc.,) this is reasonable. We would incorporate multi-category training as part of our future work.
>
> -Sec. 4 Comments on Table 3
> This is partially a result of the evaluation metric. Accuracy and Complete metrics favor different aspects. With noise in the input, the resulting mesh tends to be more “thick”, hence provides better completeness, while sacrificing the tightness of the prediction. Chamfer distance, which is the mean of the two, is resulting very similar with and without noise. We could expect the results to favor the noiseless when the noise to signal ratio is too large.

---

### Public Comment · (anonymous) · 2019-05-14
**is there any code available for this paper?**

Dear Authors,

Is there any code for this paper? I tried to find it in the GitHub but I couldn't. Are you planning to release the code to replicate the experiments?

Thanks

---

### Meta-Review · Area_Chair1 · 2018-12-13
**novel approach to interesting and challenging problem with promising results**

**Confidence:** 3
**Recommendation:** Accept (Poster)

**Metareview:**

1. Describe the strengths of the paper.  As pointed out by the reviewers and based on your expert opinion.

- The paper tackles an interesting and challenging problem with a novel approach.
- The method gives improves improved performance for the surface reconstruction task.

2. Describe the weaknesses of the paper. As pointed out by the reviewers and based on your expert opinion. Be sure to indicate which weaknesses are seen as salient for the decision (i.e., potential critical flaws), as opposed to weaknesses that the authors can likely fix in a revision.

The paper
- lacks clarity in some areas
- doesn't sufficiently explain the trade-offs between performing all computations in the spectral domain vs the spatial domain.

3. Discuss any major points of contention. As raised by the authors or reviewers in the discussion, and how these might have influenced the decision. If the authors provide a rebuttal to a potential reviewer concern, it’s a good idea to acknowledge this and note whether it influenced the final decision or not. This makes sure that author responses are addressed adequately.

Reviewers had a divergent set of concerns. After the rebuttal, the remaining concerns were:
- the significance of the performance improvements. The AC believes that the quantitative and qualitative results in Table 3 and Figures 5 and 6 show significant improvements with respect to two recent methods.
- a feeling that the proposed method could have been more efficient if more computations were done in the spectral domain. This is a fair point but should be considered as suggestions for improvement and future work rather than grounds for rejection in the AC's view.

4. If consensus was reached, say so. Otherwise, explain what the source of reviewer disagreement was and why the decision on the paper aligns with one set of reviewers or another.

The reviewers did not reach a consensus. The final decision is aligned with the more positive reviewer, AR1, because AR1 was more confident in his/her review and because of the additional reasons stated in the previous section.